# TRS: a method for determining transcript termini from RNAtag-seq sequencing data

Amir Bar[1], Liron Argaman[1], Michal Eldar[1] & Hanah Margalit [1]✉

In bacteria, determination of the 3' termini of transcripts plays an essential role in regulation of gene expression, affecting the functionality and stability of the transcript. Several experimental approaches were developed to identify the 3' termini of transcripts, however, these were applied only to a limited number of bacteria and growth conditions. Here we present a straightforward approach to identify 3' termini from widely available RNA-seq data without the need for additional experiments. Our approach relies on the observation that the RNAtag-seq sequencing protocol results in overabundance of reads mapped to transcript 3' termini. We present TRS (Termini by Read Starts), a computational pipeline exploiting this property to identify 3' termini in RNAtag-seq data, and show that the identified 3' termini are highly reliable. Since RNAtag-seq data are widely available for many bacteria and growth conditions, our approach paves the way for studying bacterial transcription termination in an unprecedented scope.

To adapt to environmental changes, bacteria strictly regulate their gene expression by versatile regulatory mechanisms at both the transcriptional and post-transcriptional levels. A fundamental step in gene expression is the determination of the transcript 5' and 3' boundaries. In addition to the extensively-studied transcription initiation regulation by transcription factors and the determination of the transcription start site, there are several mechanisms shaping the 3' termini of transcripts, including transcription termination and cleavage by ribonucleases, which affect the transcript stability and functionality. Indeed, in parallel to development of transcriptome-wide methods to determine transcription initiation sites (e.g., references[1,2]), experimental RNA-seq-based approaches have been recently developed to globally identify transcript 3' ends in bacteria. These include term-seq[3], SEnd-seq[4] and others[5,6]. In term-seq, a 3' adapter is ligated to the RNA 3' terminus prior to the RNA fragmentation, and thus, the sequences adjacent to the adapter sequence in the library reads represent original 3' termini that were present in the RNA sample[3]. SEnd-seq is based on circularization of the RNA. The sequenced reads capture the 5' and 3' ends of the RNA simultaneously, thus allowing to map both the transcription start and end sites[4]. Application of these methodologies to several bacteria, including the model organism *Escherichia coli* K-12, successfully identified the 3' termini of transcripts at a global scale[3,4,7,8].

These studies also discovered functional elements generated by premature transcription termination, including riboregulators (riboswitches or attenuators) that respond to multiple environmental signals, such as antibiotics[3,5,7]. These global studies required some manipulation of the conventional RNA-seq protocols to guarantee capturing of the 3' termini.

Recently, we and others[9] observed a distinctive accumulation of reads at transcript 3' termini in sequencing data generated by the RNAtag-seq protocol[10] (Fig. 1a). This special read pattern stems from the protocol steps, which involve random fragmentation of the RNA, followed by the ligation of an adapter (corresponding to read 1) to the 3' ends of the fragments (Fig. 1b). This results in a sequencing library in which the start of read 1 corresponds to the 3' end of a RNA fragment. For each transcript, the random fragmentation is expected to generate random fragments, but always one of them will carry the original 3' terminus of the transcript (Fig. 1c; Jonathan Livny, personal communication). Therefore, it is expected to find more reads starting at the genuine transcript 3' terminus than at the ends of other fragments produced by the random fragmentation, suggesting that recognition of this characteristic read pattern in RNAtag-seq data can be exploited for determining transcript termini, without the need for tailored experimental manipulations. To this end, we developed TRS (Termini

[1]Department of Microbiology and Molecular Genetics IMRIC, Faculty of Medicine, The Hebrew University of Jerusalem, Jerusalem 9112102, Israel.
✉e-mail: hanahm@ekmd.huji.ac.il

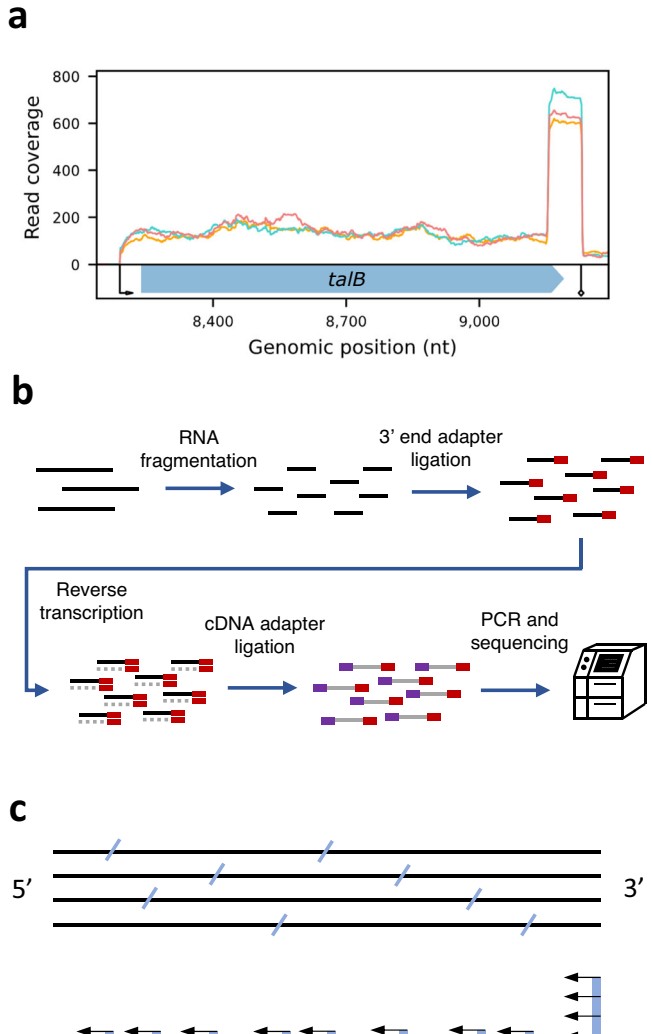

**Fig. 1 | A typical read coverage at 3' termini in RNAtag-seq data. a** Shown is the read coverage along the gene *talB* for three RNAtag-seq libraries (orange, red and cyan lines) mapped to *E. coli* K-12 MG1655 reference genome (NC_000913.3). The transcription start and termination sites (determined by SEnd-seq[4]) are marked by an arrow or a diamond arrow, respectively. The gene coding sequence is marked by a wide arrow containing the gene name. **b** Schematic representation of the RNAtag-seq protocol[10]. Briefly, the protocol involves the following steps: Random fragmentation of the RNA (black lines), RNA 3' end adapter ligation, reverse transcription (gray lines), cDNA adapter ligation, PCR and sequencing. **c** For multiple transcripts of the same RNA (black lines), break positions created by the random fragmentation (blue slashes) result in randomly distributed 3' termini except for the genuine 3' terminus, which is always present. Consequently, the number of read starts (blue bars) at the genuine 3' terminus is higher than at other 3' termini, underlying the observed pattern of reads (black arrows).

by Read Starts), a straightforward approach to identify this typical read pattern in RNAtag-seq data. We demonstrate the applicability of TRS to accurately infer the RNA 3' termini directly from sequencing data. Since RNAtag-seq data are widely available for many bacteria grown under various conditions (e.g., references[9,11–13]), TRS paves the way to study transcription termination at an unprecedented scope.

## Results

### Coverage of RNAtag-seq reads peaks at transcript 3' termini
To substantiate the conjecture that the read pattern observed at 3' termini in RNAtag-seq data results from the steps of random fragmentation followed by the 3' adapter ligation in the RNAtag-seq

protocol, we computationally simulated the RNAtag-seq library preparation process for 10,000 transcript copies of a specific gene (Methods). These transcripts were subjected to random fragmentation and a read was assigned to each fragment. Finally, we examined the read coverage along the gene, obtaining coverage per position. In accord with real data, we find in the simulations an accumulation of reads at the genuine 3' terminus (Supplementary Fig. 1), supporting the conjecture that in the RNAtag-seq actual data read peaks are expected at 3' termini. This suggests that identification of such peaks in RNAtag-seq data can be used for identification of 3' termini.

### Identification of 3' termini in RNAtag-seq data by TRS
The major steps of TRS are outlined in Fig. 2 and the detailed algorithm is described in the Methods section and in the Supplementary Information. Briefly, in single-end sequencing, the region covered by each read is determined by the position where the read start is mapped to the genome (corresponding to the 3' end of the RNA fragment), and by the read length, which often corresponds to the size defined for the sequencing machine. Therefore, the identification of read coverage peaks is equivalent to the detection of read start peaks in the RNAtag-seq data (Fig. 1c). For each library we map the reads to the genome, record the number of read starts per genomic position, and model it by the negative binomial distribution. Since 3' termini have high number of read starts mapped to them compared to the rest of the genome, the task of identifying these positions can be resolved by peak detection. However, due to variations between the libraries, applying a peak-calling algorithm separately to each library may result in inconsistent peak positions, which would need to be integrated. In TRS, prior to peak calling, we first integrate the information of multiple replicates. To this end, we developed a statistic $R_{i,j}$, that is based on the local readthrough at position i in library j, and is defined such that positions with low local readthrough (putative 3' termini) get high values. This statistic considers the number of read starts in a small window around a position ($L_{i,j}$) and the number of read starts in a window downstream ($D_{i,j}$). The statistic scales read start counts to values between 0 and 1 and its expected value can be utilized to set a threshold (T), above which a position is considered a putative 3' terminus (Methods). In practice, we compute for each position $i$ the average value of $R_{i,j}$ across the libraries and apply a peak-calling procedure to these average values, keeping for further analysis positions of average $\bar{R}_i$ peaks with values $\geq$ T. We then test for each individual library whether the read start counts at the determined $\bar{R}_i$ peak positions differ statistically significantly from the read start counts downstream to them, and record statistically significant positions that recur in multiple libraries. Finally, we classify the 3' termini by gene and transcript annotations (Fig. 3, Methods).

### 3' termini detected by TRS applied to RNAtag-seq data are highly reliable
Initially, we applied TRS to three sequencing libraries of published RNAtag-seq data generated in our lab for RNA extracted from *E. coli* K-12 MG1655 cells grown on LB to exponential phase[14]. In these data we identified a total of 1486 3' termini (Supplementary Data 1-2). For assessment, we compared our results with data of 3' termini detected experimentally by methods dedicated to this aim: two studies that used term-seq[7,8] and one study that used SEnd-seq[4] (Fig. 4). Our set of 3' termini had a statistically significant overlap with each of the three datasets of 3' termini ($p \leq$ 1E-467 by hypergeometric test; Methods), and the levels of overlap were comparable to those obtained when comparing the results for each pair of these datasets (Supplementary Fig. 2). Of note, while there are statistically significant overlaps between the different datasets, each dataset has a large set of uniquely identified 3' termini (Fig. 4). This might be due to differences in the library preparation procedures (term-seq, SEnd-seq, or RNAtag-seq) and in the computational pipelines, as well as to variations in the

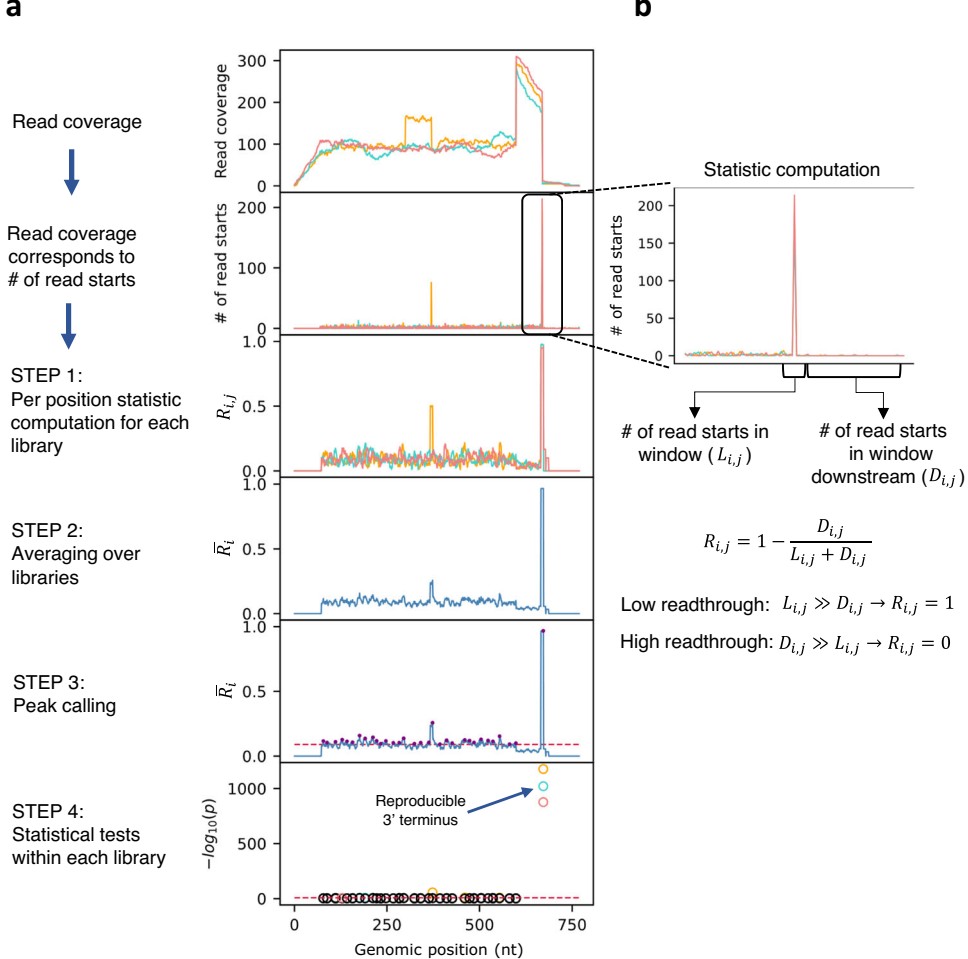

**Fig. 2 | Schematic presentation of the TRS algorithm. a** Read coverage pattern corresponds to the number of read starts in each library (reflecting the 3′ termini of RNA fragments). The number of read starts per position is the input of the algorithm. Step 1: We compute the statistic $R_{i,j}$, which measures the local readthrough per genomic position and scales the libraries to the same range of values. Step 2: The averages of $R_{i,j}$ across libraries ($\bar{R}_i$) are computed. Step 3: We apply a peak-calling procedure to $\bar{R}_i$ values and determine putative 3′ termini (purple dots) above a preset threshold (red dashed line). Step 4: For each library, using the original read start counts, we apply a statistical test to the peak positions identified in Step 3 (Methods). The *p*-values are corrected for multiple hypothesis testing. Positions with *p*-value ≤ 0.01 (red dashed line) in multiple libraries are determined as 3′ termini. Corrected *p*-values are presented by -log₁₀(p). Corrected *p*-values of statistically significant 3′ termini in each library are shown as circles, colored by the library color, and otherwise in black. **b** $R_{i,j}$ corresponds to the local readthrough, computed for each library. It is defined as the ratio between the number of reads that pass the position (i.e., reads that start downstream to the position and counted by $D_{i,j}$) and the total number of reads covering it (i.e., reads that start downstream or reads that start at the position counted by $D_{i,j} + L_{i,j}$). Presented is the number of read starts in a region around the statistically significant peak shown in **a**. The number of read starts at the local region is high compared to the downstream region (i.e., $L_{i,j} \gg D_{i,j}$) and hence $R_{i,j}$ approaches 1.

bacterial strains or in the RNA samples due to possible differences in the growth conditions or RNA extraction methods.

A most adequate assessment of the 3′ termini detected by the application of TRS to RNAtag-seq data can be achieved by comparing them to 3′ termini obtained by an independent methodology applied to the same RNA samples, analyzed by the same computational method. While TRS was originally developed to determine 3′ termini using RNAtag-seq data, we realized that this approach is general enough to analyze other datasets with 3′ termini enriched signals. Indeed, when we applied TRS to the Dar and Sorek term-seq data[8], we found a statistically significant overlap between the 3′ termini identified by TRS and those reported in the original paper (Supplementary Information). Thus, we can apply TRS to term-seq and RNAtag-seq data generated for the same RNA samples, in order to obtain a most accurate assessment of the reliability of 3′ termini based on RNAtag-seq data.

To this end, we have extracted total RNA from *E. coli* K-12 MG1655 grown to exponential phase (OD₆₀₀ of ~0.4) in either rich (LB) or minimal (EG) medium (three replicates each), and used the extracted RNA for construction of RNAtag-seq and term-seq libraries (Methods). Reads were then mapped to the reference genome and 3′ termini positions were determined by applying TRS to the data generated by each methodology (Methods, Supplementary Table 1, Supplementary Data 3), resulting in four datasets of 3′ termini: (i) LB RNAtag-seq, (ii) EG RNAtag-seq, (iii) LB term-seq, and (iv) EG term-seq. We describe in the text the comparison of the 3′ termini determined in the LB datasets (*i* and *iii*), and in the Supplementary Information the comparison of the results for the EG datasets (*ii* and *iv*). We identified 1814 3′ termini in the LB RNAtag-seq data and 1984 3′ termini in the LB term-seq data. 1316 of these 3′ termini were co-discovered in the data generated by both methods, a highly statistically significant overlap ($p \leq 1E-1387$ by hypergeometric test; Fig. 5a; Methods). The 3′ termini determined by both methods in the EG datasets were also highly consistent (Supplementary Information; Supplementary Fig. 3).

There were 498 3′ termini unique to the LB RNAtag-seq dataset, which were not identified in the LB term-seq dataset (Fig. 5a).

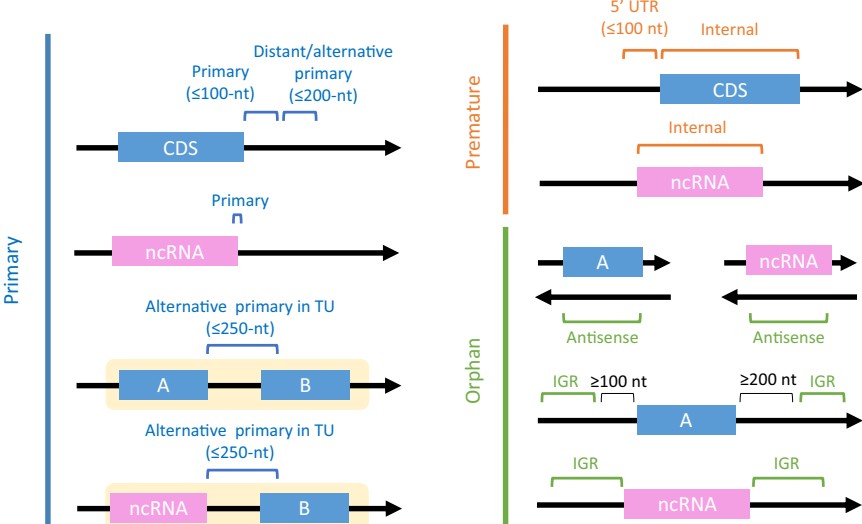

**Fig. 3 | Classification of 3′ termini according to gene and transcript annotations.** In all panels, blue rectangles represent the protein coding sequence (CDS) of a gene, pink rectangles represent non-coding RNA (ncRNA) genes, and black arrows represent the transcription orientation. Genes that reside in the same transcription unit are surrounded by yellow shading. 3′ termini are divided into three groups and subgroups: (1) Primary – 3′ termini located downstream the stop codon of protein coding genes or at the end of ncRNA genes. This group includes: (i) Primary 3′ termini, positioned up to 100 nucleotides downstream the stop codon of a protein coding gene or at the end of a ncRNA gene. (ii) Distant Primary (DP) termini, positioned up to 200 nucleotides downstream the stop codon of a protein coding gene, when there is no 3′ terminus in the first 100 nucleotides. (iii) Alternative Primary (AP) termini, same as distant primary termini but there is a 3′ terminus in the first 100 nucleotides. (iv) Alternative Primary termini in Transcription Unit (AP in TU), assigned to genes that are not last in their operon (either CDS or ncRNA gene). In this case the region downstream is extended up to 250 nucleotides. (2) Premature – 3′ termini located within the 5′ UTR of genes or in their CDS or in a ncRNA gene. (3) Orphan – 3′ termini located antisense to genes (AS) or in intergenic regions distant from genes (IGR).

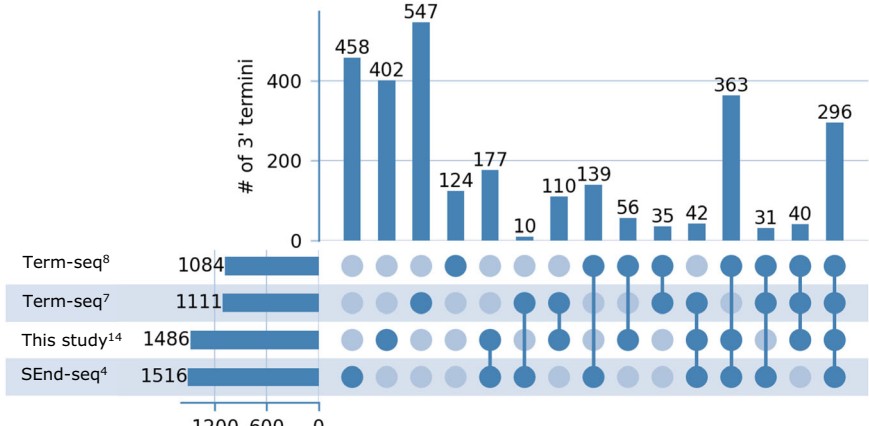

**Fig. 4 | Overlap of 3′ termini detected by TRS applied to RNAtag-seq data with previously published 3′ termini datasets.** Comparison of the set of 3′ termini detected by applying TRS to published RNAtag-seq data[14] and previously published 3′ termini datasets obtained by term-seq[7,8] and SEnd-seq[4]. Each row represents a dataset, and each column represents the intersection of the 3′ termini in corresponding datasets (dark blue circles). The number of 3′ termini in each dataset or intersection is presented by horizontal and vertical bars, respectively.

To further assess the validity of these 3′ termini, we extracted their flanking sequences, aligned them at the 3′ terminus positions and analyzed the positional nucleotide distributions. Indeed, we observed nucleotide enrichment matching a GC-rich terminator hairpin structure followed by a poly uridine tail, similar to the motif identified for common 3′ termini detected in both RNAtag-seq and term-seq datasets (Supplementary Fig. 4a). In addition, the predicted free energy values of the regions upstream the 3′ termini are comparable to the predicted free energy values of shared 3′ termini, identified in both RNAtag-seq and term-seq data (Supplementary Fig. 4b, Supplementary Information). These results suggest that most of the 3′ termini unique to the RNAtag-seq dataset are true termini that were not detected by term-seq. Further analysis has revealed that 68% of the RNAtag-seq unique termini were not identified in the term-seq data because they had low read coverage in at least two term-seq libraries (below our minimal threshold). The other 32% of 3′ termini not detected by term-seq had sufficient coverage but they did not pass the statistical test (Fig. 5a). Notably, 60% (297/498) of the 3′ termini identified in the LB RNAtag-seq data but not in LB term-seq data were discovered in the EG term-seq data or in the previous studies mentioned above[4,7,8]. Taken together, compiling the 3′ termini discovered in the LB RNAtag-seq data that are supported by at least one of the above datasets, we can estimate the precision of the 3′ termini identified in the RNAtag-seq data to be at least 89%. Classifying the 3′ termini by gene and transcript annotations (Fig. 5b, c), we observe that for primary 3′ termini (at the end of fully-transcribed genes or operons), the RNAtag-seq precision rises to 96% (Table 1).

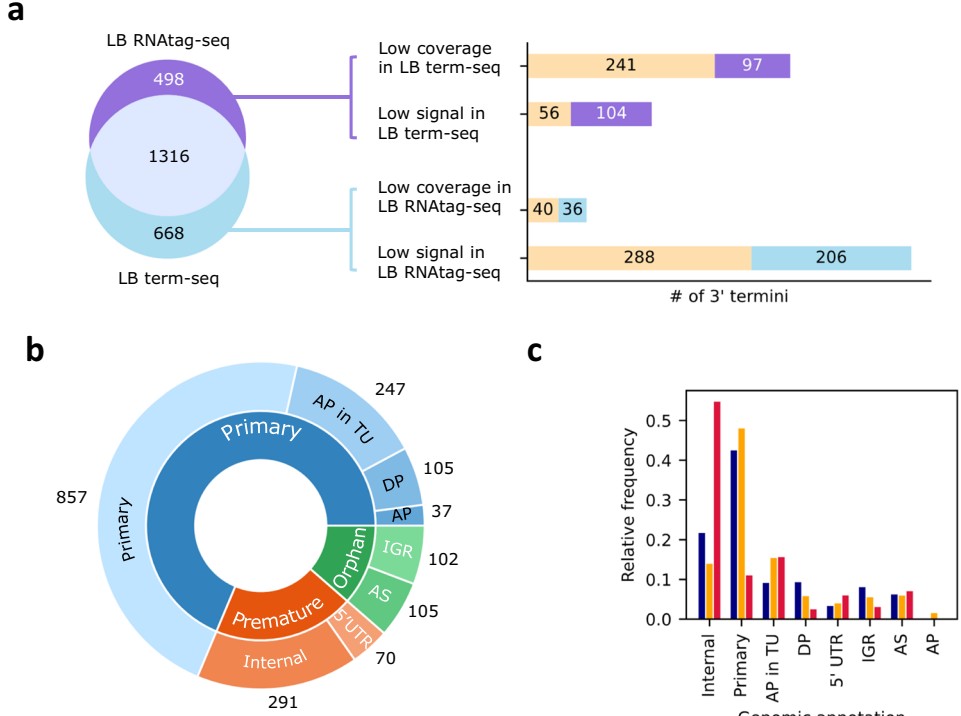

**Fig. 5 | Assessment of 3' termini detected by TRS applied to RNAtag-seq data.** **a** Comparison between the 3' termini identified by TRS applied to data of RNAtag-seq and term-seq conducted on the same RNA samples from cells grown in rich (LB) medium. 3' termini unique to each protocol were further analyzed for the possible reason they were not detected by the other protocol, either due to low coverage or due to statistically insignificant *p*-value (low signal). The classification of 3' termini into these two categories are presented as colored bars, where the number of 3' termini supported by previous studies[4,7,8] or in our EG term-seq libraries are marked in yellow. **b** Classification of 3' termini identified in the LB RNAtag-seq dataset. The 3' termini annotations are divided into eight classes following Fig. 3 (outer circle), which can be categorized by the super classes: primary, premature, and orphan 3' termini (inner circle). **c** Categories of 3' termini identified by the various methods. 3' termini obtained by applying TRS to RNAtag-seq and term-seq data were divided into three groups: overlapping 3' termini (orange bars), 3' termini unique to RNAtag-seq (blue) or term-seq (red) datasets. Shown is the relative frequency of each 3' terminus category within its group.

In parallel, 66% of all 3' termini identified in the term-seq data are identified in the RNAtag-seq data. This percentage rises to 82% when focusing on primary 3' termini, implying also high sensitivity for these sites. Other classes of 3' termini showed lower overlap, especially internal 3' termini residing within coding sequences, or 3' termini residing in intergenic regions of operons. While for some of the internal 3' termini the read start signals completely vanished in the RNAtag-seq data, for others (85%) a signal was observed but it was too weak to pass our statistical test (Fig. 6). The algorithm performance on RNAtag-seq data is summarized in Table 1, showing that read start peaks in RNAtag-seq data can be exploited for the determination of genuine 3' RNA termini with high precision and high sensitivity.

There are potential biases in the reported results, associated with various steps of the RNAtag-seq protocol. One potential bias may arise from the ligation step in case the ligase has a preference for distinct nucleotides. To assess possible ligase preferences, we aligned the sequences at the positions of mapped read starts within CDSs, and looked at the nucleotide enrichment in the flanking regions

(Supplementary Fig. 4c). There was no nucleotide bias at the ligation point. Interestingly, we identified a slight enrichment of A at the second read position, as well as a slight enrichment of U at the genomic position downstream the mapped read start, possibly associated with some fragmentation bias. Overall, the difference between this motif and the identified terminator motif (Supplementary Fig. 4a) suggests that the 3' termini identified by TRS are not affected by this slight bias or by any ligase preference. Another potential bias may originate from the PCR step amplifying the same cDNA fragment multiple times (known as PCR duplicates). In duplication events, the number of reads starting at the duplicated fragment position might be over-represented compared to other positions and it could be falsely identified as a 3' terminus. To assess the effect of possible PCR duplicates we analyzed paired-end RNAtag-seq libraries[15,16], from which PCR duplicates can be removed. To this end, we removed all but one read of identical sequences and applied TRS to the filtered read data. Excluding 3' termini known from previous studies[4,7,8], we compared the detected 3' termini in the filtered data to those detected in the original data (see Supplementary Information). The results of this analysis indicated that the fractions of 3' termini in the original data that might have been falsely detected due to PCR duplicates (Supplementary Fig. 5) change among the 3' termini associated with various genomic annotations, from -15% (premature 3' termini) to -2% (primary 3' termini). In paired-end data, where duplicates can be removed, this artefact can be addressed.

Finally, we verified that TRS identifies 3' termini in RNAtag-seq datasets published by other research groups for *E. coli*[7,15,17,18]. Comparative analysis of *E. coli* primary 3' termini identified in other RNAtag-seq datasets have shown high consistency with the previously

## Table 1 | TRS performance

|  | All 3' termini | | Primary 3' termini | |
|---|---|---|---|---|
|  | Precision | Sensitivity | Precision | Sensitivity |
| Based on LB term-seq alone | 1316/1814 (73%) | 1316/1984 (66%) | 651/862 (76%) | 651/728 (89%) |
| Based on LB term-seq and additional datasets[4,7,8] | 1613/1814 (89%) | – | 841/876 (96%) | – |

**a**

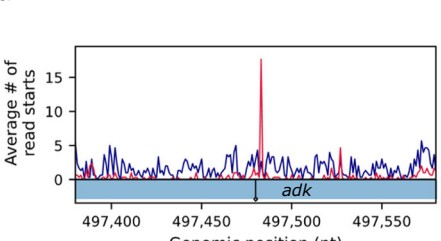

**b**

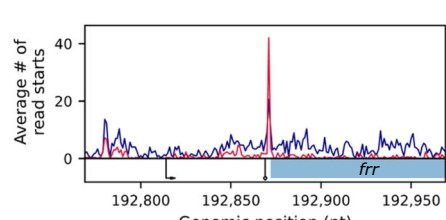

**Fig. 6 | Examples of 3' termini identified in the term-seq dataset but not in the RNAtag-seq dataset.** Shown are read start patterns of RNAtag-seq (blue) and term-seq (red). **a** 3' terminus of *adk*, identified within the CDS of the gene. Read starts accumulate in term-seq data but not in RNAtag-seq data. **b** 3' terminus identified within *frr* 5' UTR. Read starts do accumulate in the RNAtag-seq data but not to the same extent as in the term-seq data. The y-axes of **a** and **b** are not scaled. The gene coding sequences are marked by blue rectangles below the read coverage plots. Arrows are as in Fig. 1a.

identified ones (Supplementary Information and Supplementary Fig. 6). Furthermore, for two bacterial species we had access to both RNAtag-seq and term-seq datasets from other labs: *E. coli* K-12[7] and *Listeria monocytogenes*[3,11]. For each of the bacterial species there was high consistency between the determined 3' termini in the two corresponding datasets (Supplementary Information and Supplementary Fig. 7a–c). Finally, we analyzed RNAtag-seq data of three other bacteria, *Klebsiella pneumoniae*, *Salmonella enterica* and *Shigella flexneri*[11], and demonstrate the repertoires of identified 3' termini along with the 3' terminus repertoires of *E. coli* K-12 and *L. monocytogenes* (Supplementary Information and Supplementary Fig. 7d).

### Insights gained from the 3' termini identified in RNAtag-seq data by TRS

Having vast data of RNAtag-seq of different bacteria grown under various conditions (e.g., references[11,19]), along with the ability to identify 3' termini by solely exploiting these data, opens the door to study questions of posttranscriptional regulation at a transcriptome-wide scale. The mere data of RNAtag-seq of a bacterium grown under a specific condition should enable the discovery of regulatory elements embedded in the RNA by identification of internal 3' termini. Having RNAtag-seq data of a bacterium grown under different conditions enables comparison of the 3' termini of genes between conditions and detection of possible switches between primary and internal 3' termini, which might indicate on premature transcription termination or processing under a certain condition, acting as regulatory mechanisms. Having RNAtag-seq data of various bacteria should enable a comparative study looking at the conservation of such mechanisms, as well as at the degree of conservation of primary 3' termini. Here we provide examples for each of these types of insight, which can be gained by applying TRS to RNAtag-seq data:

1. 3' UTR-derived small RNAs - Using the 3' termini map we obtained for *E. coli* K-12 MG1655, we can accurately define the 3' UTR of transcripts and measure their expression by computing the average number of read starts along the 3' UTR per library. Likewise, we can assess the expression within the coding sequence (CDS) by computing the average number of read starts along these regions. Examining these values in mRNAs of protein coding genes that are transcribed alone or last in their operon, we observed that the average numbers of read starts within the CDS and 3' UTR of genes are highly correlated ($r = 0.84$–$0.85$ for the different libraries, $p \leq 1.91E{-}238$, Fig. 7a, Supplementary Fig. 8). Yet, we identified 38 outlier transcripts (Fig. 7a, Supplementary Data 4, Methods). Out of these, five genes had a higher average number of read starts in the CDS compared to the 3' UTR. For two of these genes (*pmrR* and *mgtT*), SEnd-seq showed transcription termination of the upstream genes inside the CDSs, while two other genes (*yoeH* and *bax*) may have undergone premature termination or processing within their own sequences. 34 genes had a higher number

of read starts in the 3' UTR compared to the CDS, hinting at putative RNAs derived from the 3' UTR. These include the 3' UTR of *malM* (Fig. 7b), shown previously to be stabilized by binding to ProQ and involved in many interactions with other RNAs[15], the 3' UTR of *tdcG* (Fig. 7c), which was found in a previous study to interact with 20 different transcripts when bound to Hfq[19,20], and several other transcripts presenting extremely low expression levels in the coding sequence, yet relatively high expression levels at their 3' UTRs. Among these candidates is a short RNA embedded in *chiQ* 3' UTR. Interestingly, in exponential phase, *chiQ* is known to be completely downregulated through premature transcription termination enhanced by the interaction of the sRNA ChiX with the 5' UTR of *chiP*, preceding *chiQ* in an operon[21]. Yet, we do observe high coverage in *chiQ* 3' UTR (Fig. 7d), suggesting a short RNA may be produced from an internal transcription start site, putatively providing an additional layer in the regulation of chitosugar metabolism. Alternatively, since *chiQ* shares a terminator with *fur*, which is encoded on the complementary strand, the putative transcript derived from *chiQ* 3' UTR might serve as an antisense to *fur* 3' UTR. Overall, our results show that putative 3' UTR-derived sRNAs may be inferred by integrating expression levels and 3' termini determined by TRS, both based on the same RNAtag-seq data. Furthermore, this analysis emphasizes the advantage in integrating different layers of information from the same data, leading to insights that could not be obtained by analyzing each information layer alone.

2. Identification of condition-specific termini - Comparative analysis of 3' termini between growth conditions may unravel termination and processing events that are more abundant under one condition compared to another condition. This mainly regards a primary 3' terminus identified under one condition and a premature 3' terminus identified under the other condition. It is possible that both termini are identified under both conditions, but they may differ in their magnitude, possibly indicating that the premature 3' terminus serves for regulation, for example, for maintaining the desired amount of the full-length mRNA and encoded protein. To assess the magnitude of a premature 3' terminus we used the measure based on the readthrough at those positions, defined above as $\bar{R}_i$. Based on the properties of $\bar{R}_i$ (Methods), it approaches 0 for full readthrough and is expected to be relatively high in cases of 3' termini, including premature termination or processing. Comparative analysis of the transcription readthrough measures obtained for various growth conditions may reveal putative condition-dependent premature transcription termination or processing (Fig. 8a). A similar logic was previously used with term-seq data to successfully determine condition-specific regulators[3], with the limitation that RNA-seq data had to be generated for genes within operons with alternative promoters. In our case, RNAtag-seq directly provides these two layers of information, the 3' termini data and full gene coverage data that enable us to estimate the local readthrough.

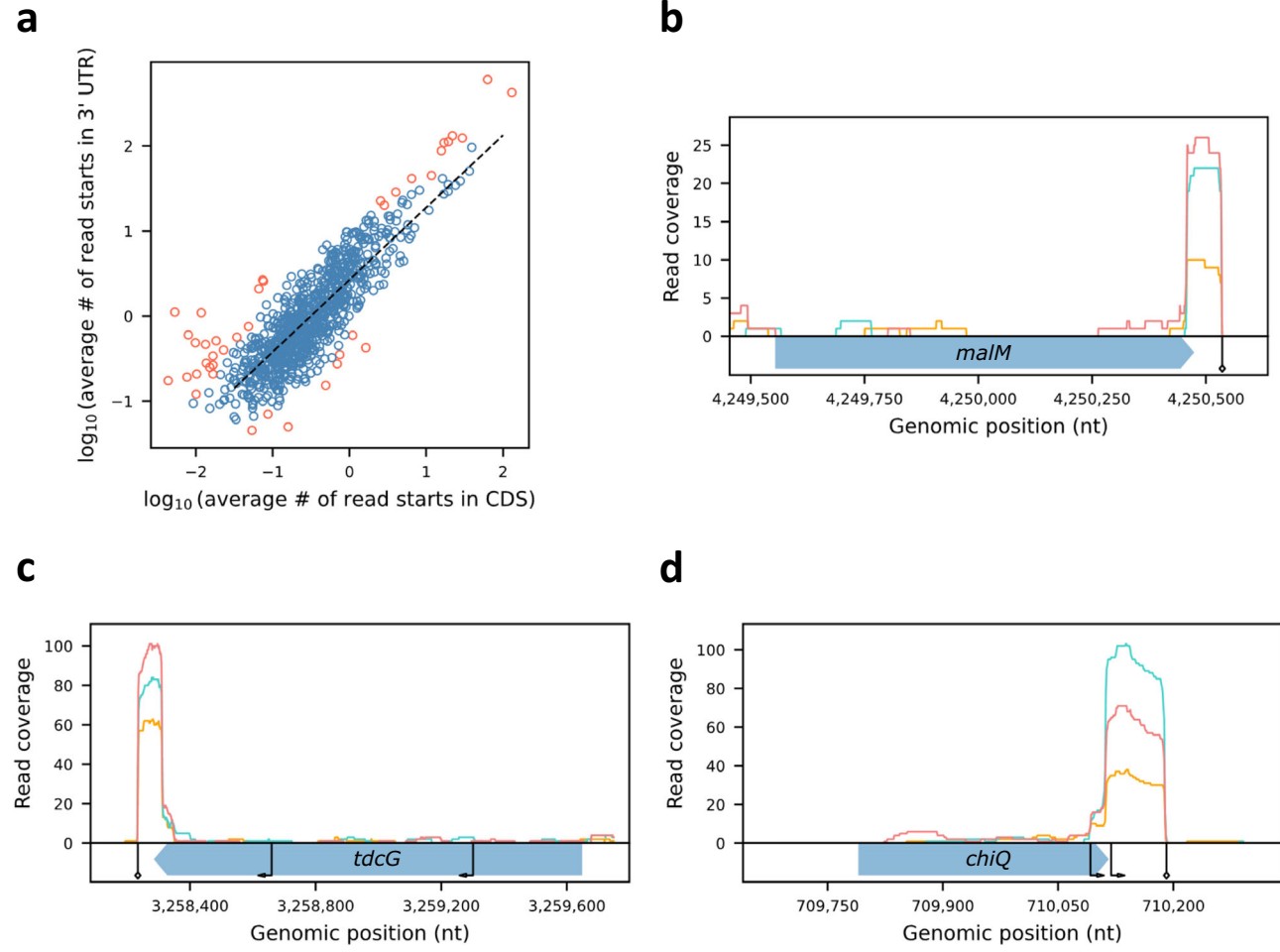

**Fig. 7 | 3' UTR-derived transcripts identified in the LB RNAtag-seq dataset. a** For each gene with a primary or a distant primary 3' terminus in the LB RNAtag-seq dataset, the $\log_{10}$ transformed average number of read starts within the CDS and 3' UTR were computed. Presented is the scatterplot of these values for one of the libraries and the regression line fitted (dashed black line). The correlation coefficient is r = 0.85 ($p \leq 3.58E{-}252$ by two-sided Student's *t* test). Results for the other two libraries are presented in Supplementary Fig. 8. Genes that were identified as outliers (Methods) are colored red. **b**–**d** Presented is the coverage along the genes *malM* (**b**), *tdcG* (**c**), and *chiQ* (**d**) that were identified as outliers in **a**. Transcription start sites identified by Thomason et al.[43] are indicated by arrows and the 3' termini by diamond arrows. The gene coding sequences are marked by wide arrows containing the gene names.

To assess the ability of TRS to discover condition-specific termination, we applied it to previously published RNAtag-seq data of enteropathogenic *E. coli* (EPEC), generated in our lab[19]. In those experiments, EPEC was grown to two growth phases, either to stationary phase (on LB medium) or to exponential phase (on DMEM, which mimics infection conditions). Hence, we could use these RNA-seq data to search for condition-specific termini (Supplementary Data 5). In this analysis we considered 3' termini with sufficient read coverage (at least 20 reads per position in at least two replicates per condition). We further narrowed down this list by excluding rRNA genes and including only termini identified as premature, resulting in a list of 133 candidates. We expect higher abundance of short RNA fragments due to premature termination or processing in the growth phase/condition where the measure $\bar{R}_i$ is higher compared to the other growth phase/condition. We computed the difference in $\bar{R}_i$ measures between the conditions, ranked the genes by the absolute values of these differences, and selected genes above the median for experimental testing following manual visualization of their read profiles in the two conditions. We experimentally tested by northern blot analysis three genes in which we identified growth phase/condition-dependent premature 3' termini (*uspA*, *rpsL*, and *rpsA*), and verified shorter transcripts for two of them in stationary phase/LB (Fig. 8b, c). For *uspA* we did not identify in the northern blot analysis a shorter RNA fragment

despite its strong indication by the readthrough measure obtained for the stationary phase/LB RNAtag-seq data (Fig. 8b). Our experimental results support premature termination or processing at stationary growth phase/LB for the two other genes, *rpsA* and *rpsL*, which encode for ribosomal proteins S1 and S12, respectively (Fig. 8b, c). It is well established that there is a decrease in the translation of proteins during stationary phase. Reduction in the level of ribosomal proteins, which are usually highly expressed[22], is one way to achieve this. Indeed, it is known that the translation of the ribosomal protein S1, encoded by *rpsA*, is auto-regulated post-transcriptionally by S1 binding to its own mRNA[23]. The premature transcription termination or processing that we discovered at stationary phase may add another layer to the regulation of this gene. Of note, the RNA size implied by the northern blot analysis is shorter than the size suggested by the premature termination, hinting at possible further upstream processing. A similar logic may apply to *rpsL*, which is regulated post-transcriptionally by the ribosomal protein S7[24], sharing an operon with S12. The premature termination or processing might join this regulation to reduce S12 levels at stationary phase.

3. Conservation of RNA regulatory elements in bacteria - The vast amounts of RNAtag-seq data available for diverse bacteria provide an opportunity to study whether termination sites of genes are conserved among different bacteria. Computing the distance between the stop

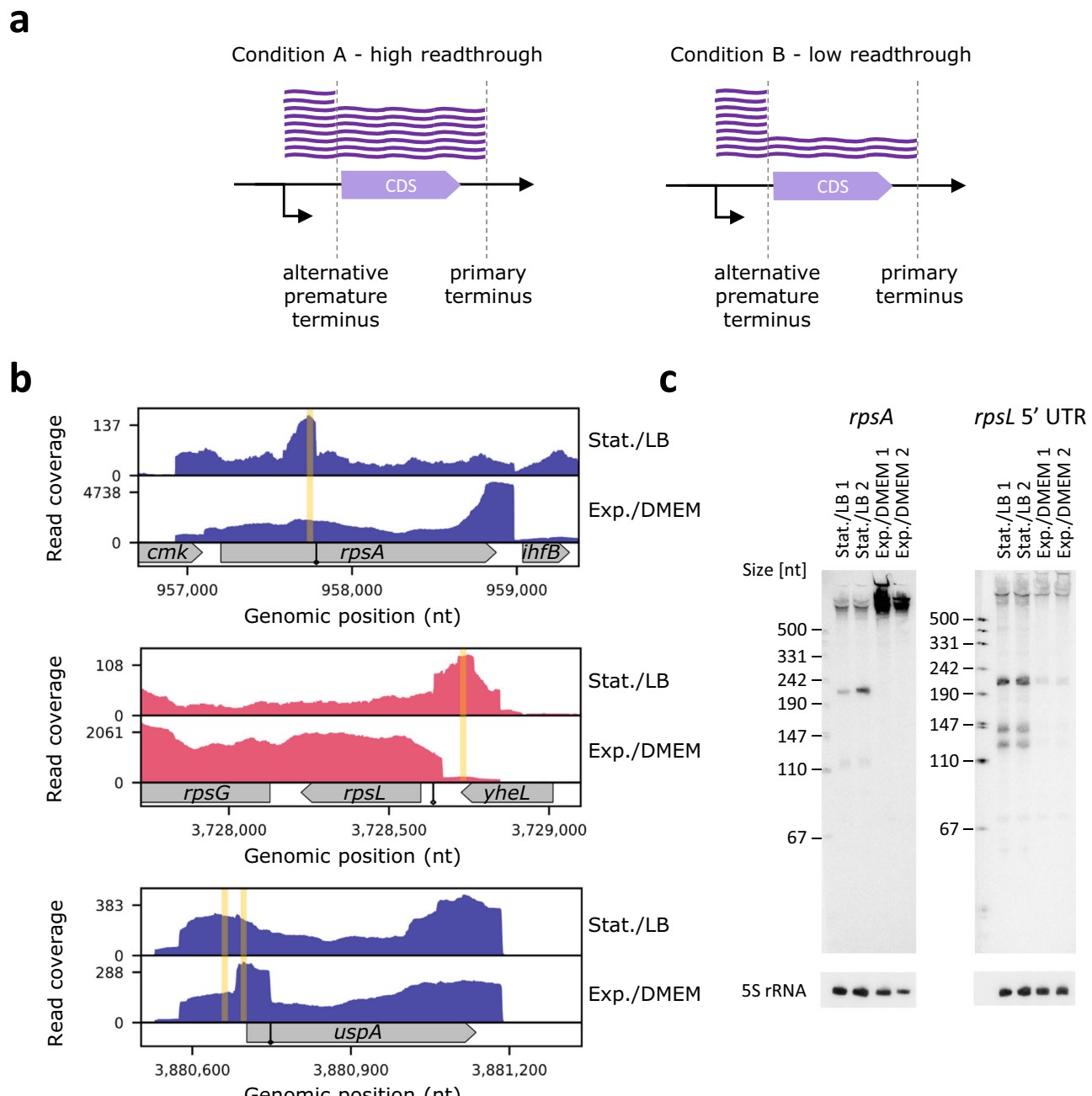

**Fig. 8 | Examples of conditional 3' termini in EPEC. a** Schematic representation of a primary 3' terminus (left panel) that changes to a premature 3' terminus (right panel) in response to change in conditions. Under one condition (left panel) most transcripts extend through the alternative premature 3' terminus and end at the primary 3' terminus, manifesting high readthrough at the premature 3' terminus. Under another condition (right panel) most transcripts end at the premature termination site (low readthrough). **b** RNAtag-seq read coverage around the conditional 3' termini (marked by a diamond arrow) in stationary phase/LB and exponential phase/ DMEM conditions. Presented are 3' termini unique to the stationary phase/LB medium for *rpsA*, *rpsL*, and *uspA*. Highlighted in yellow are the probe locations designed for the northern analysis described in **c**. **c** Verification of conditional 3' termini in *rpsA* and *rpsL* by northern analysis. Total RNA extracted from EPEC cultures grown to stationary phase on LB or to exponential phase on DMEM was analyzed, using gene specific probes. 5S rRNA was probed as a loading control. The experiment was done with two biological repeats.

codon and a 3' terminus assigned to a gene can provide a comparable measure of a 3' terminus position between different bacteria. Positive values indicate 3' terminus positions that are downstream the stop codon, usually determining the length of 3' UTR. Negative values indicate 3' terminus positions that are upstream the stop codon, which may imply premature termination or processing. By this measure, similar distances indicate conservation and dissimilar distances indicate a change in the termination site[25]. For example, conservation of 3' termini located within 5' UTRs, may hint at the conservation of

regulatory elements generated by premature termination or processing in various bacteria.

Applying this computation to 3' termini determined by TRS applied to RNAtag-seq data of *Enterotoxigenic E. coli* (ETEC)*, Salmonella enterica* Typhimurium, *Klebsiella pneumoniae*, and *Shigella flexneri*[11], we found examples of regulatory elements that are conserved across these bacterial pathogens. For example, the expression of the Mg$^{2+}$ transporter *mgtA* is regulated by MgtL, a leader peptide that is encoded upstream to *mgtA* in the operon. Under conditions

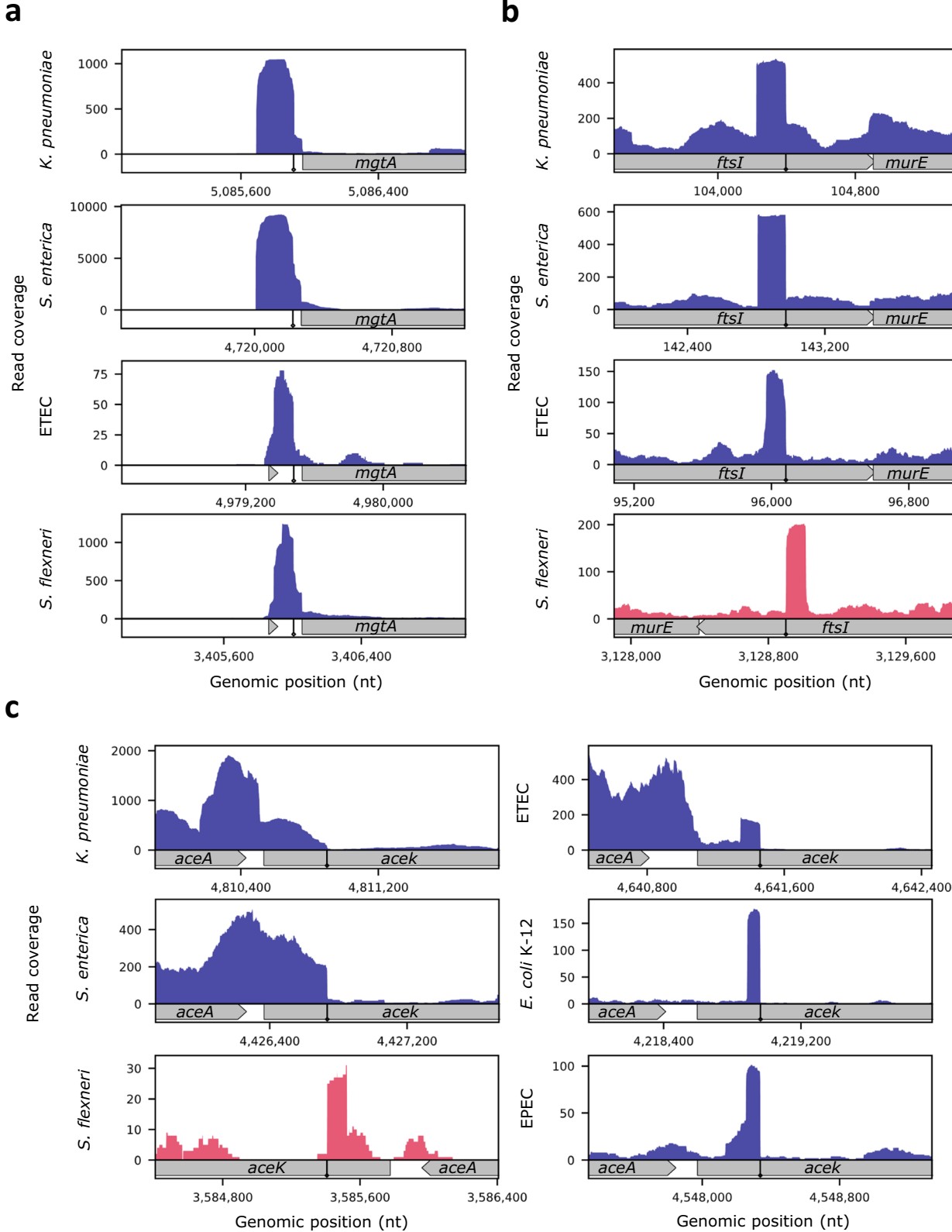

where $Mg^{2+}$ is abundant, the translation of *mgtL* is increased and enables the formation of a stem-loop structure that results in premature termination of *mgtA*[26–28]. Interestingly, in all the bacteria listed above we identified a 3′ terminus at similar positions in all orthologous genes, indicating that premature termination of *mgtA* is conserved (Fig. 9a). Another example is the small RNA FtsO generated from the *ftsI* gene transcript, which was identified in *E. coli* K-12[7]. While it was

shown that its sequence is conserved in other bacteria, it was not demonstrated that the short RNA FtsO has been generated in the other species. Here we show that an internal 3′ terminus in *ftsI* is found in multiple bacteria at the exact same position of orthologous genes (Fig. 9b). Another example is a sRNA derived from the CDS is AceK-int, a ~85-nt long sRNA derived from the *aceK* gene[7,16] (Fig. 9c). We were able to verify there is a 3′ terminus at the same relative position in all

**Fig. 9 | Conserved 3′ termini of regulatory elements and sRNAs in bacteria.**
Presented is the read coverage around conserved 3′ termini of a regulatory element and sRNAs in *E. coli* K-12 and four other bacteria: *K. pneumoniae*, *S. enterica*, ETEC, and *S. flexneri*. The 3′ terminus (marked by a diamond arrow) was determined by applying TRS to previously published RNAtag-seq data for the different bacteria[11]. **a** A premature 3′ terminus identified upstream to *mgtA* CDS. **b** The 3′ terminus matching the sRNA FtsO in *E. coli* K-12[7]. **c** The 3′ terminus matching the sRNA AceK-int in *E. coli* K-12, encoded within the CDS of *aceK*[7,16]. Presented 3′ termini are based on data of growth conditions that showed the highest read coverage and exhibited

the most consistent results across the different bacteria. The growth conditions per gene and bacterium are as follows: *mgtA* – *S. enterica*, *K. pneumoniae*, and *S. flexneri* (acidic stress), ETEC (nutritional downshift). FtsO – *S. enterica*, *S. flexneri*, and ETEC (control), *K. pneumoniae* (heat shock). *aceK* – *S. enterica*, *K. pneumoniae*, and ETEC (nutritional downshift), *S. flexneri* (acidic stress), EPEC (stationary phase), *E. coli* K-12 (exponential phase). The read coverages of *K. pneumoniae*, *S. enterica* and EPEC are based on paired-end sequencing, and for ETEC, *S. flexneri*, and *E. coli* K-12 on single-end sequencing.

*aceK* orthologous genes, however, not in all bacteria the sRNA AceK-int is apparent. In EPEC paired-end RNAtag-seq data we do see read coverage matching the AceK-int sRNA of *E. coli* K-12[19]. Yet, in *K. pneumoniae* and *S. enterica* paired-end RNAtag-seq data there is a longer transcript that does not seem to match AceK-int and could be the 3′ UTR of the upstream gene *aceA*. As to ETEC and *S. flexneri* single-end RNAtag-seq data, although it looks as if there is an independent transcript, in fact the reads extend to a length greater than 85 nt, hinting that also in these bacteria the sRNA is absent. In addition, sequence alignment by Clustal Omega[29] from the start codon of *aceK* up to the internal termini in K-12, ETEC, and EPEC indicates that the K-12 and ETEC are identical and the EPEC sequence is of ~95% identity. This suggests that sequence identity does not guarantee the generation of the sRNA and other factors may be involved as well. In addition, it is possible that the read patterns observed in the various bacteria provide hints to the evolution of AceK-int.

## Discussion

We present a novel approach to analyze RNA-seq data, using the read patterns rather than read counts. Similar analyses of peaks in read patterns were applied before to data generated by tailored applications of RNA-seq, such as term-seq[7] and 3pMap[5]. However, RNA-seq data per se were mainly analyzed for detection of differentially expressed genes by comparing read counts. Here we demonstrate that RNA-seq data may hold additional layers of information that represent molecular mechanisms involving the RNA, which by innovative analyses may expand our understanding of basic mechanisms, as we exemplified here for transcription termination. Previous studies in other contexts also demonstrated that non-conventional analysis of RNA-seq data may reveal novel molecular mechanisms and regulatory molecules, such as the discovery of circular RNA[30,31].

By applying TRS to RNAtag-seq data of *E. coli* grown to exponential phase, we showed that 3′ termini obtained by this analysis are highly compatible with 3′ termini obtained by other approaches. Furthermore, strict analysis of 3′ termini obtained by TRS applied to RNAtag-seq and term-seq data of the same RNA samples showed that 3′ termini identified by term-seq are rediscovered in the RNAtag-seq data, and for primary 3′ termini our results are with high precision and sensitivity. Yet, our method is less sensitive for internal termini, many of which are determined in term-seq data but not in RNAtag-seq data (Fig. 5c). In essence, the RNAtag-seq protocol could be considered as a noisy version of term-seq for identifying 3′ termini. This stems from the random fragmentation preceding the 3′ adapter ligation in the RNAtag-seq protocol, which generates additional fragments with 3′ termini, compared to term-seq. Accordingly, many internal term-seq peaks appear as less substantial using the RNAtag-seq data. Nonetheless, the significant advantage of our approach is in the availability of vast amounts of RNAtag-seq data for many bacterial species, providing an opportunity to study transcription termination in wide scope.

The access to ample sequencing data enabled us to compare 3′ termini between growth conditions and between bacteria. Utilizing the data in different conditions, we identified 3′ termini residing at the 5′ UTR or CDS in which the readthrough measure changed between conditions, hinting at regulatory mechanisms manifested by premature transcription termination or processing. Comparative analysis

of the corresponding 3′ termini identified in different bacteria revealed high conservation between bacteria (Fig. 9). This can supplement sequence conservation analysis by providing experimental evidence of a consistent 3′ terminus in different bacteria. A repeatedly identified premature 3′ terminus in different bacteria can potentially serve as an indicator for an important regulatory event. Nonetheless, results based on such analyses should be further thoroughly investigated, as corresponding 3′ termini may have different roles in different bacteria, as is hinted by the AceK-int example (Fig. 9c).

Along with transcription termination, processing of RNAs by ribonucleases is a major mechanism for generating 3′ termini in the cell. Despite that, while we can associate many 3′ termini to transcription termination sites of genes, we could attribute only a small number of 3′ termini to endoribonuclease cleavage sites. In our data (RNAtag-seq and term-seq), only a minor portion of the identified 3′ termini overlapped with known endoribonuclease cleavage sites (~5% for RNase III[32] and 2% for RNase E[33], see Supplementary Information). This may suggest that most cleavage products with newly generated 3′ termini are further degraded, possibly by 3′ to 5′ exonucleases, which either result in shorter stabilized transcripts or are completely degraded.

Similar to other large-scale analyses, TRS is susceptible to technical biases associated with the experimental procedure. First, since the RNAtag-seq protocol involves a ligation step, there may be a bias towards specific sequences. While other protocols for the identification of 3′ termini of transcripts also involve a ligation step (i.e., term-seq and 3pMap), RNAtag-seq is more prone to this artifact since RNA fragments are sheared prior to the ligation step, generating a larger repertoire of fragments. However, our analysis of nucleotide enrichments around mapped read start positions did not reveal such a bias (Supplementary Fig. 4c). Second, another potential bias may arise due to PCR duplicates. In our assessments, we provided estimations of the fractions of 3′ termini that might have been falsely detected due to PCR duplicates (overall, 8–9%). These are actually upper bounds for the putative fractions of false detections, as some of the reads might represent true biological RNA copies. While we showed this putative false detection is minor for primary 3′ termini, we recommend that when conducting single-end sequencing one should aim at minimizing the number of PCR cycles and take more caution when observing a premature 3′ terminus. Recall that paired-end data can be exploited to identify and remove the PCR duplicates.

As mentioned above, determination of the 3′ termini in RNAtag-seq data relies on the ligation of the adapter corresponding to read 1 to the 3′ end of the RNA fragments. Conceivably, ligation of the adapter corresponding to read 1 to the 5′ end of the RNA fragments is expected to result in read start peaks at 5′ termini. This suggests that RNA-seq data of libraries that involve adapter ligation at 5′ ends can be exploited to determine the transcription start sites of RNAs, without the need in special experiments designed to this aim, like dRNA-seq[2]. Paired-end sequencing data can provide information on both the 5′ and 3′ termini of transcripts, directly from the sequencing data, by analyzing the read starts of the two reads separately (Supplementary Fig. 9).

The ability to annotate the transcript termini directly from sequencing data is highly beneficial. First, it would be very useful for

precise counting of reads that are mapped to genes in the RNA sequencing experiment. Currently, in the absence of precise transcript boundaries, the gene read counts are determined based on the coding sequence boundaries, missing the untranslated parts of the transcript that might contain important information, as we show (Fig. 7). Furthermore, as mentioned above, many studies used the RNAtag-seq protocol for RNA sequencing, and ample sequencing data generated by this protocol are available. The ability to determine the termini based on this vast amount of data opens the door to wide-range studies of the conservation of the termini between conditions and between orthologous genes in different bacterial species. Such analyses can provide insights into molecular mechanisms that control the positions of the termini, as well as condition- or species-dependent regulation of transcription termination.

## Methods

### RNAtag-seq and term-seq experiments
Cultures of *E. coli* MG1655 were grown while shaken at 37 °C in either rich LB medium or minimal E medium pH 7.0[34] supplied with 0.4% glucose (EG medium). At $OD_{600}$ of ~0.4 the cells were collected, and total RNA was extracted using TRI-reagent (Sigma T9424). RNAtag-seq libraries were constructed as described[35]. Term-seq libraries were constructed according to Dar et al.[3], with slight changes. 1 μg of TURBO DNase (Invitrogen AM1907) treated total RNA was ligated to a 3′ barcoded adapter (150 pmole) using 54 units of T4 RNA ligase 1 (New England Biolabs M0204), in a 20 μl reaction at 22 °C. Ligation reaction was arrested after two hours by addition of 60 μl RLT buffer (Qiagen 79216). The adapter ligated RNA samples were pooled together and cleaned-up using the Zymo Clean and Concentrator™-5 kit. The RNA was then fragmented using Ambion Fragmentation Reagent (Ambion AM8740) and cleaned-up using 2.5 volumes of RNAClean XP beads (Beckman-Coulter) and 1.5 volumes of isopropanol, according to manufacturer instructions. The RNA was depleted of ribosomal RNA using Ribo-Zero (Illumina), cleaned-up using 2.5 volumes of RNAClean XP beads (Beckman-Coulter) and 1.5 volumes of isopropanol and then used for cDNA synthesis using Invitrogen SuperScript™ III First-Strand Synthesis System. The cDNA was cleaned-up twice with 1.5 volumes of Ampure XP (Beckman-Coulter) and amplified by 9 cycles of PCR. RNAtag-seq and term-seq libraries were sequenced by 85 cycles of single-end sequencing using NextSeq500 instrument. The sequences of oligonucleotides used are listed in Supplementary Table 2.

### Northern blots
Cultures of enteropathogenic *E. coli* (EPEC) E2348/69 were grown statically (without shaking) at 37 °C over-night. The next day, the cultures were diluted 1:50 in DMEM (Biological industries, Cat #01-053-1A) and grown statically at 37 °C to $OD_{600}$ of ~0.3. Total RNA was extracted from both the over-night cultures (denoted 'LB' samples) and from the DMEM cultures, using TRI-reagent (Sigma T9424). 15 ug of total RNA were separated on an acrylamide-urea gel [6% acrylamide/bisacrylamide 19:1 (Biolabs 000-135233500), 7 M urea (Supelco 1.08487.1000)] in TBE buffer, and transferred onto a Zeta-Probe blotting membrane (Bio-Rad 1620159). The membranes were hybridized with specific 5′-end radiolabeled probes in ULTRAhyb-Oligo Hybridization Buffer (Invitrogen AM8663) and visualized by Phosphorimager (Typhoon FLA 7000, GE Life Sciences). Probe sequences are listed in Supplementary Table 2.

### Mapping of sequenced reads
Sequencing adapters were removed using cutadapt[36] version 3.4 (-m 25 −q 15 −a AGATCGGAAGAGC −n 5 −e 0.15 −j 0), and processed reads were mapped to *E. coli* K-12 MG1655 reference genome (NC_000913.3) using bwa[37] version 0.7.17-r1188 (bwa aln −n 2 −t 8 −R 200 followed by bwa samse).

### Simulation of the RNAtag-seq protocol
In the RNAtag-seq protocol, RNA is extracted, randomly fragmented (by hydrolysis after heating the RNA) and a barcode is ligated at its 3′ end. To simulate the read pattern generated by the RNAtag-seq protocol, we simulated the transcript fragmentation process, assigned sequencing reads to the 3′ regions of the fragments, and analyzed the read coverage pattern. The random fragmentation was simulated by sampling N positions from a uniform distribution (the selection of N is as described[38]). Next, we trimmed the RNA fragments from the 3′ ends to the maximal sequencing read length (70 nucleotides), and treated these fragments as reads that are then mapped to the genome. Since our computational pipeline for mapping reads to the genome requires a minimal read length of 25 nucleotides, we filtered fragments shorter than 25 nucleotides. Finally, for each position, we recorded the number of reads mapped to it.

### Algorithm for detecting 3′ termini in RNAtag-seq data
**Model description.** We denote by $X_{i,j}$ the number of reads of library j that start at the $i^{th}$ position. Commonly, count data, such as $X_{i,j}$, is modeled by a Poisson distribution that is characterized by a single parameter for its mean and variance. This approach was used, for example, in the 3pMap algorithm for determining 3′ termini[5]. Alternatively, due to the over-dispersion of such data, in some applications (e.g., DESeq2[39] and edgeR[40]) it was modeled by the negative binomial distribution, in which the mean and the variance differ. Since our data is over-dispersed (Supplementary Fig. 10), we use the negative binomial distribution to model it.

Previous studies already showed that 3′ termini generated by either transcription termination or processing do not necessarily occur at an exact single position and can span a few nucleotides upstream or downstream the major site[8,32]. To capture this property, we define a random variable, $L_{i,j}$, which is the count of read starts in library j spanning a window of 2 W positions around the $i^{th}$ position, $L_{i,j} = \sum_{k=i-W}^{i+W} X_{k,j}$. Under the assumption that $X_{i-W,j}, \ldots, X_{i+W,j}$ are independent and identically distributed, the distribution of $L_{i,j}$ is also negative binomial and its parameters are governed by W and $X_{i,j}$'s distribution parameters. Intuitively, we would expect the region downstream 3′ termini to have substantially fewer read starts, as in the case of processing, and almost none at strong transcription termination sites. Thus, to capture this property, we define a variable, which is the number of read starts in a region starting at position i + W + 1 and spanning D positions downstream, $D_{i,j} = \sum_{k=i+W+1}^{i+W+D} X_{k,j}$, which has also a negative binomial distribution.

**Narrowing down putative 3′ termini.** To integrate the information of multiple replicates, read start counts per position should be normalized. Here, we introduce the statistic $R_{i,j} = \frac{L_{i,j}}{L_{i,j}+D_{i,j}} = 1 - \frac{D_{i,j}}{L_{i,j}+D_{i,j}}$, which scales the data and essentially measures the level of transcription readthrough (or, actually, 1-"readthrough"). The statistic has the following properties:

(i)   It is in the range [0,1] and hence, on the same scale for all replicates.

(ii)  When there is no readthrough (as expected at transcription termination sites), i.e., when $D_{i,j} = 0$, then $R_{i,j} = 1$.

(iii) When there are relatively many read starts downstream the considered position (i.e., $D_{i,j} \gg L_{i,j}$), $R_{i,j}$ approaches zero.

(iv)  When there is no difference in the distributions of $L_{i,j}$ and $D_{i,j}$ (for example, within an unprocessed coding region), the mean of the statistic is independent of the region coverage and depends only on the sizes of the local region and the region downstream (see Supplementary Information and Supplementary Fig. 11).

Properties (i) and (iv) imply that taking the statistic average value $\bar{R}_i = \frac{1}{M} \sum_{j=1}^{M} R_{i,j}$ across M replicates is meaningful. For positions that are not genuine 3′ termini, it is expected to result in values around the

expected mean of $\bar{R}_i$, while positions of genuine 3′ termini would deviate from it. Furthermore, property (iv) enables us to set a threshold for the minimal value required for a position to be considered a peak. In practice, we found that the procedure of transforming read start counts to the statistic, averaging across replicates, and determining positions above the expected mean is useful in reducing the number of tested positions while maintaining positions of known 3′ termini.

**Selection of positions with statistically significant peaks.** The number of read starts in the positions determined by the above analysis are next subjected to statistical testing. For the statistical tests, we return to consider read start counts and look at each library separately (Supplementary Information). Our null hypothesis is that the tested peak does not represent a 3′ terminus and hence there is no difference between the distribution of read start counts per position around the tested position and in the region downstream (the positions considered for computing $L_{i,j}$ and $D_{i,j}$, respectively). It follows that we can estimate the distribution parameters of $L_{i,j}$ from the region downstream (Supplementary Information). Now, we can compute the probability to get a number of read starts around the position (L) that is greater than or equal to the value we observed ($L_{i,j}$), and test whether it is lower than or equal to some significance threshold ($\alpha$) under the null hypothesis. In other words, we can test whether $P(L \geq L_{i,j}) \leq \alpha$ where L follows the negative binomial distribution with the estimated parameters. Finally, since we assess many positions, we apply Bonferroni correction for multiple hypothesis testing.

**Final set of 3′ termini.** After applying all the steps above, we obtain a table with N rows (the number of putative positions) and M columns (the number of replicates) containing the p-value per position for each replicate. Positions that passed the statistical test for all M replicates are considered as 3′ termini. However, as this requirement may be too strict, the pipeline enables to relax this requirement by selecting a minimal number of statistically significant replicates in addition to setting a minimal threshold for the statistic $\bar{R}_i$. Since a ribosomal RNA depletion step [Ribo-Zero (Illumina)] was applied in the library construction, which may affect read distribution at rRNAs, these were not included in the final dataset. In addition, following previous studies for determining 3′ termini[4,7,8] we excluded tRNAs as well (after verifying we identified all tRNAs).

**Parameter values used for TRS analysis**
3′ termini were determined by running our algorithm using a local window of size 3 nucleotides (W = 3 nucleotides), downstream region of 67 nucleotides (D = 67 nucleotides), and a stringent statistical significance threshold of 0.01. We included in our final dataset 3′ termini that were either statistically significant in all three libraries or statistically significant in two libraries with $\bar{R}_i \geq 0.5$. The parameters W and D were determined as described in the Supplementary Information, and the threshold for $\bar{R}_i$ was determined based on its distribution (Supplementary Fig. 12). In addition, we have set a minimal threshold of 10 reads per position, otherwise, $R_{i,j}$ was set to 0. For the peak-calling procedure, we required a minimal distance of 10 nucleotides between two neighboring peaks. Although the computational pipeline computes the minimal threshold of $\bar{R}_i$ of the peak calling procedure to be 0.095, for the above parameters we manually rounded it to 0.1. This value was still below the mean of the statistic plus one standard deviation (Supplementary Fig. 11).

TRS was applied with the above parameters to all datasets in this study. It is of note that some datasets included only two libraries, and for those, the 3′ termini had to be identified in all replicates. Naturally, in such cases of only two replicates, the sequencing depth may strongly affect the number of identified 3′ termini (Supplementary

Fig. 13a). Hence, although using two libraries is also possible, we recommend using three libraries. Larger number of libraries were not tested in this study and parameter adjustments may be required.

**Annotation of 3′ termini**
Each 3′ terminus was assigned one of eight different classes according to its genomic position relative to genes, transcription units, terminators, and promoters reported in the EcoCyc 25.5 database[41] (Fig. 3). These classes of 3′ termini include: (A) Primary 3′ termini, divided to four categories: (i) Primary – assigned only to 3′ termini related to genes that are transcribed independently or when the gene is the last in all transcription units it is part of. A 3′ terminus was set as primary if it was within the 3′ UTR of a gene with a known terminator, or if it was at most 100 nucleotides downstream the stop codon of a protein coding gene. For non-protein-coding RNAs, the documented 3′ terminus was used as Primary. (ii) Alternative Primary (AP) – assigned to 3′ termini located at most 100 nucleotides downstream primary transcript regions of CDS, when there was a primary terminus identified. (iii) Distant Primary (DP) – same as alternative primary, however, applies when there was no primary 3′ terminus identified. (iv) Alternative Primary in Transcription Unit (AP in TU) – assigned to 3′ termini related to genes that are not last in any transcription unit up to 250 nucleotides downstream the stop codon of a protein coding gene, but not exceeding the last position in the operon. (B) Premature 3′ termini, divided to two categories: (v) Internal – assigned to 3′ termini that occur within the CDS of protein coding genes or prematurely in non-coding RNAs. (vi) Premature in 5′ UTR – assigned to 3′ termini located within the 5′ UTR of protein coding genes. When the 5′ UTR is unknown, it is estimated to at most 100 nucleotides upstream the initiation codon. (C) Orphan 3′ termini, divided to two categories: (vii) Orphan in antisense (AS) – assigned to 3′ termini in antisense to a non-coding RNA or to the CDS of a protein coding gene. (viii) Orphan in intergenic regions (IGR) – assigned to 3′ termini within intergenic regions that were not assigned any of the above. In case of ambiguity in classification, the order of classes is as follows: (iv), (i), (v), (vi), (ii), (iii), (vii) and (viii). For each putative terminus, a window of five nucleotides is considered when assessing it in view of the annotation.

**Sequencing data collection**
All sequencing libraries were either downloaded manually or using the SRA-toolkit program fasterq-dump version 2.9.3 (https://trace.ncbi.nlm.nih.gov/Traces/sra/sra.cgi?view=software) as fastq or bam files. See Supplementary Data 1 for the accession numbers of the various datasets.

**Compilation of 3′ termini from previous studies**
In this study, we compared the set of 3′ termini identified by TRS applied to RNAtag-seq data and 3′ termini identified by term-seq[7,8], and SEnd-seq[4]. While all these studies provide a table specifying the 3′ termini identified, there are some differences that require pre-processing before applying a direct comparison. First, in Dar and Sorek[8] the *E. coli* BW25113 strain was used while *E. coli* K-12 MG1655 strain was used in the rest. Second, all datasets filtered tRNA and rRNA from further analysis, but the methodology used to remove these was not reported. Here, to ensure that 3′ termini that may be related to rRNA and tRNA are completely removed, we filtered all 3′ termini located within 100 nucleotides upstream or downstream the seven rRNA operons (as defined in EcoCyc 25.5[41]) and tRNA genes.

To transform the 3′ termini coordinates from the BW25113 strain (CP009273) genome to the *E. coli* K-12 MG1655 (NC_000913.3) strain genome, we first aligned the two genomes using Mugsy[42] (version 1r2.3) to map overlapping regions. Then, BW25113 3′ termini that are found within overlapping regions were transformed to their corresponding positions in the *E. coli* K-12 MG1655 genome.

## Computation of overlapping 3' termini between datasets and statistical significance

For our analyses, we considered a pair of 3' termini overlapping if they were located at most 10 nucleotides apart. To compute whether the number of overlapping termini was statistically significant, we used a hypergeometric test with N – total number of positions in the genome, K – number of 3' terminus positions in the reference dataset along with flanking 10 nucleotides upstream and downstream, n – number of identified 3' termini, k – number of overlapping 3' termini.

## Identification of outliers in 3' UTR – CDS regression model of read start counts

In the analysis we modeled the dependency between the average counts of read starts across the positions in the CDS and the 3' UTR of protein-coding genes by an ordinary least squares regression. Subsequently, we computed the Cook's distance for each gene and the average Cook's distance across all genes. Genes that had a (distance/average distance) >3 in at least two libraries were considered outliers.

## Reporting summary

Further information on research design is available in the Nature Portfolio Reporting Summary linked to this article.

## Data availability

The RNAtag-seq and term-seq sequencing libraries generated in this study have been deposited in the ArrayExpress database under accession code E-MTAB-12429. All referenced sequencing libraries accession codes and their respective reference genome used in this study are listed in Supplementary Data 1. Figures in this study contain information on transcription start sites from previously published datasets (Thomason et al.[43] and Ju et al.[4]). This study used in some of the analyses previously published data of 3' termini (term-seq[7,8] and SEnd-seq[4]) and cleavage sites (RNase III[32] and RNase E[33]). The data supporting the findings of this study are available from the corresponding authors upon request. Source data for the figures and supplementary figures are provided as a Source Data file. Source data are provided with this paper.

## Code availability

The implementation of the TRS algorithm is available as a python package in the Python Package Index and GitHub (https://github.com/amirbarHUJI/TRS).

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

## Acknowledgements

This study was supported by the European Research Council Advanced Grant #833598 (H.M). and the Israeli Science Foundation Grant # 1253/22 (H.M) A.B. is partially supported by a fellowship of the Data Sciences program of the Planning and Budgeting Committee of the Israel Council for Higher Education. We thank Johnathan Livny for insightful discussions, Y. Altuvia for fruitful discussions and valuable comments on the manuscript, Y. Gatt-Harpaz for his assistance in the alignment of *E. coli* genomes and his very useful comments, I. Warshavsky for his assistance in mapping additional sequencing datasets, and T. Kaplan and G. Elidan for their useful advice.

## Author contributions

Initiated and supervised the study – H.M.; Experimental Investigation – L.A.; TRS Algorithm and Software development – A.B.; Computational Analysis – A.B. and M.E. Writing – H.M., A.B. and L.A.

## Competing interests

The authors declare no competing interests.
