## [Peer Review File · Nature Communications]

TRS: a method for determining transcript termini from RNAtag-seq sequencing dataREVIEWER COMMENTS

Reviewer #1 (Remarks to the Author):

The authors describe an analytical methods to re-purpose the data derived from RNAtag-seq sequencing for the identification of 3' termini in bacteria. They also experimentally compare the result of the pipeline on RNAtag-seq and term-seq performed on the same sample. This study is sound and useful as it provide a mean to add an additional set of information on top of gene expression analysis.

Major comments :

1- The title “TRS: a method for determining transcript termini from sequencing data” is somehow misleading to believe that any sequencing will do - nonetheless, only the sequence derived from RNAtag-seq protocol can be used for the identification of termini. The authors may want to modify the title to reflect this fact.

2- Randomness of fragmentation and ligation steps : It is known that ligation has biases that may result in favoring some sequence contexts over others. In this case, false positive termination sites may be resulting from an accumulation of reads in a particularly favorable ligation spot. To address this concern, the authors could for example combine all the termini that are specific to TRS and evaluate, using sequence logo for example, wether a specific context is found.

3- Library complexity. Can the authors comments on the fact that some of the RNAtag-seq libraries may contains a large number of PCR duplicates which will notably affect the ability of the algorithm in calling termination sites. This is a point that can be either added as a warning in the discussion or the authors can evaluate what levels of PCR duplicates are acceptable using simulation.

4- RNAtag-seq does not distinguish between transcription termini and process termini, The last paragraph of the supplemental material attempt to find process termini but maybe the authors could, in addition ,add this point to the discussion for clarification.

Minor comments:

Figure 1A : genomic position : is it in bp ? Which reference genome has been used ? Same question for all figures with genomic locations.

<https://github.com/amirbarHUJI/TRS#demo-example> which reference genome has been used ?

Discussion : “Here we demonstrate, yet again, that RNA-seq data may hold additional layers of information” can the authors reference previous literatures that shows that RNA-seq hold additional

layers of information ?

Not sure I get the statement : “the terminal fragment will always carry the original 3’ terminus of the transcript” Sounds more like a tautology to me.

“Initially, we applied TRS to published RNAtag-seq data generated in our lab for RNA extracted from E. coli K-12 MG1655 cells grown on LB to exponential phase” Can the authors provide the number of replicates for this dataset ?

I do not have access to the deposited sequences.

Reviewer #2 (Remarks to the Author):

In this article Bar et al., develop a new computational method (Termini by Read Starts) to identify 3’ ends from bulk RNA-seq datasets, when those RNA-seq libraries have been built using the RNAtag-seq. The authors document that their approach can successfully identify a substantial subset of 3’ ends, largely 3’ ends directly downstream of annotated genes, from E. coli RNAtag-seq datasets. This method is powerful in that it will provide an additional tool for researchers to apply when performing total RNA-seq analysis. The method does have limitations, which the authors are upfront about, mainly that their approach is less sensitive to detect ORF-internal 3’ ends.

Critiques:

- The authors validate their approach by comparing E. coli 3’ ends previously identified by Term-seq or SEnd-seq and by analyzing new E. coli RNAtag-seq and Term-seq datasets. Most of the analysis is focused on the later comparison, as they wanted to assess their methodology by a direct comparison of 3’ end identification using both approaches (TRS v. Term-seq) using the same RNA samples. While these data are important, it does not globally address the efficacy of TRS to identify 3’ ends from different RNAtag-seq experiments. The authors should analyze other published E. coli RNAtag-seq data, not just a comparison of their data to Term-seq/SEnd-seq analysis from other labs. Their arguments would be strengthened by analyzing several RNAtag-seq datasets from different labs by TRS and comparing 3’ ends identified to a list of previously known/identified E. coli 3’ ends. This analysis could be specifically focused on primary 3’ ends (downstream of genes), as they document their approach is most sensitive in detecting this category of 3’ ends.

- Another study has performed E. coli RNAtag-seq and Term-seq on the same RNA samples (PMID: 33460557). The authors compare 3’ ends identified by Term-seq from this study to other datasets. However, they should also analyze and directly compare their TRS method on the RNAtag-seq in this previous study to the Term-seq 3’ end identification in this previous study – as they have done with their own data (Figure 4B).

- The authors should consider how effective their approach will be for other bacteria. Term-seq has now been carried out for numerous bacteria. This study would be strengthened by analyzing RNAtag-seq data from other bacteria and reporting the efficacy of TRS to find 3' ends. Two studies have recently been submitted to BioRxiv, one for *Mycobacterium tuberculosis* (<https://doi.org/10.1101/2022.06.01.494293>) and another for *Borrelia burgdorferi* (<https://doi.org/10.1101/2023.01.04.522626>) which documented a majority of 3' ends mapped internal to ORFs. Will TRS analysis only be successful for a subset of bacterial RNAtag-seq datasets?
- It is not obvious how the last section of the results provides any new insights gained from TRS. Yes, their analysis found several long 3' UTRs from RNAtag-seq data, but these examples (Figure 5) and 3' ends have been reported in other studies, some even with northern analysis to document an sRNA. Rather this section, could speak to novel discoveries from their approach/an example for the application of TRS to find new information from existing datasets.
- The authors should comment on the number of replicates required/ideal for TRS analysis.

Reviewer #3 (Remarks to the Author):

This manuscript by Bar et al. presents a bioinformatics method named TRS to identify 3' termini of bacterial transcripts from RNAtag-seq data. The method is based on the hypothesis that real 3' termini of transcripts are significantly enriched in RNAtag-seq data as compared to those artificial termini generated by RNA fragmentation, a step used for transcript sequencing. The methodology is interesting and potentially useful in finding terminus regulation. However, the work as presented is largely a technical trick for terminus analysis. The authors should expand their work to mine existing data in the public database and present termini that are variably used in different conditions. This would make this work suitable for publication in a high profile journal like Nature Communications. In addition, RNA ligation efficiency can have substantial influences on how RNA fragments are captured and made into cDNA. Presumably, some artificial termini generated by RNA fragmentation may have a much higher ligation efficiency than true termini, leading to false positives. The authors need to address this possibility as well.

Response to Reviewers

Reviewer #1

The authors describe an analytical methods to re-purpose the data derived from RNAtag-seq sequencing for the identification of 3' termini in bacteria. They also experimentally compare the result of the pipeline on RNAtag-seq and term-seq performed on the same sample. This study is sound and useful as it provide a mean to add an additional set of information on top of gene expression analysis.

Major comments:

1- The title “TRS: a method for determining transcript termini from sequencing data” is somehow misleading to believe that any sequencing will do - nonetheless, only the sequence derived from RNAtag-seq protocol can be used for the identification of termini. The authors may want to modify the title to reflect this fact.

We changed the title accordingly and it is now

TRS: a method for determining transcript termini from RNAtag-seq data

2- Randomness of fragmentation and ligation steps: It is known that ligation has biases that may result in favoring some sequence contexts over others. In this case, false positive termination sites may be resulting from an accumulation of reads in a particularly favorable ligation spot. To address this concern, the authors could for example combine all the termini that are specific to TRS and evaluate, using sequence logo for example, whether a specific context is found.

Following the reviewer's comment, we conducted a couple of analyses: 1) We aligned the sequences at termini that were determined in the RNAtag-seq data but not in the term-seq data, and analyzed the nucleotide frequencies in flanking positions along the aligned sequences. Indeed, we identified a common sequence motif at these 3' termini, however, it matched the GC-rich sequence of a terminator hairpin structure followed by a poly-uridine tail. This common motif further supports the identified 3' termini as genuine ones. 2) To assess possible ligase preferences, we aligned the sequences at the positions of mapped read starts within CDSs, and looked at the nucleotide enrichment in the flanking regions. There was no nucleotide bias at the ligation point. Interestingly, we identified a slight enrichment of A at the second read position, as well as a slight enrichment of U at the genomic position downstream the mapped read start, possibly associated with some fragmentation bias. Overall, the difference between this motif and the identified terminator motif suggests that the 3' termini identified by TRS are not affected by this slight bias or by any ligase preference. Pages 5-6 lines 148-158, Pages 6-7 lines 178-187 and in the Supplemental Information (Supplementary Fig. 4).

3- Library complexity. Can the authors comments on the fact that some of the RNAtag-seq libraries may contains a large number of PCR duplicates which will notably affect the ability of the algorithm in calling termination sites. This is a point that can be either added as a warning

in the discussion or the authors can evaluate what levels of PCR duplicates are acceptable using simulation.

Following the reviewer's comment, we conducted an analysis to estimate the potential effect of PCR duplicates on our results. We used previously published paired-end RNAseq libraries (Melamed et al., 2020 and Bar et al., 2021), in which duplicate fragments can be identified and removed. We applied TRS twice to these data sets, before and after removing PCR duplicates, showing that their effect on the results is small for primary 3' termini. We added this analysis to the results (page 7 lines 188-199), Supplemental Information (page 10 lines 249-276), Supplementary Fig. 5) and discuss these aspects in the Discussion of the revised manuscript (page 13 lines 394-401), in a paragraph discussing the effect of experimental biases on the identified 3' termini

4- RNAseq does not distinguish between transcription termini and process termini, The last paragraph of the supplemental material attempt to find process termini but maybe the authors could, in addition ,add this point to the discussion for clarification.

Following the reviewer's comment, we added this point to the discussion (page 13 lines 380-387).

Minor comments:

- *Figure 1A: genomic position: is it in bp? Which reference genome has been used? Same question for all figures with genomic locations.*

In the revised manuscript, we added the units (nucleotides) to all figures with axes labeled "genomic position".

We also updated the description of the demo example in Github to include the reference genome that was used to map the sequencing reads.

- *Discussion: "Here we demonstrate, yet again, that RNA-seq data may hold additional layers of information" can the authors reference previous literatures that shows that RNA-seq hold additional layers of information?*

In the revised manuscript we explained in more detail a previous example concerning the discovery of circular RNA, where the read data was analyzed in an innovative way (page 12 lines 350-355).

- *Not sure I get the statement: "the terminal fragment will always carry the original 3' terminus of the transcript" Sounds more like a tautology to me.*

We rephrased the paragraph to make it clearer (page 2 lines 57-59).

- “Initially, we applied TRS to published RNAtag-seq data generated in our lab for RNA extracted from *E. coli* K-12 MG1655 cells grown on LB to exponential phase” Can the authors provide the number of replicates for this dataset?

The number of replicates was added (page 4 lines 111-113).

I do not have access to the deposited sequences.

We apologize for this inconvenience. The data is now accessible.

Reviewer #2

In this article Bar et al., develop a new computational method (Termini by Read Starts) to identify 3' ends from bulk RNA-seq datasets, when those RNA-seq libraries have been built using the RNAtag-seq. The authors document that their approach can successfully identify a substantial subset of 3' ends, largely 3' ends directly downstream of annotated genes, from E. coli RNAtag-seq datasets. This method is powerful in that it will provide an additional tool for researchers to apply when performing total RNA-seq analysis. The method does have limitations, which the authors are upfront about, mainly that their approach is less sensitive to detect ORF-internal 3' ends.

Critiques:

- *The authors validate their approach by comparing E. coli 3' ends previously identified by Term-seq or SEnd-seq and by analyzing new E. coli RNAtag-seq and Term-seq datasets. Most of the analysis is focused on the later comparison, as they wanted to assess their methodology by a direct comparison of 3' end identification using both approaches (TRS v. Term-seq) using the same RNA samples. While these data are important, it does not globally address the efficacy of TRS to identify 3' ends from different RNAtag-seq experiments. The authors should analyze other published E. coli RNAtag-seq data, not just a comparison of their data to Term-seq/SEnd-seq analysis from other labs. Their arguments would be strengthened by analyzing several RNAtag-seq datasets from different labs by TRS and comparing 3' ends identified to a list of previously known/identified E. coli 3' ends. This analysis could be specifically focused on primary 3' ends (downstream of genes), as they document their approach is most sensitive in detecting this category of 3' ends.*

In the revised manuscript we have added analyses of *E. coli* RNAtag-seq datasets published by four different labs. We conducted comparative analyses between the primary 3' termini determined by TRS in each of these datasets and previously published primary 3' termini determined by the methods term-seq and SEnd-seq (i.e., the 3' termini used as reference). We show a statistically significant overlap between primary 3' termini determined by TRS applied to these RNAtag-seq datasets and the reference primary 3' termini. These results are described in

the main text (page 7 lines 200-203) and in the Supplemental Information (page 7-8 lines 175-191, Supplementary Figure 6).

- *Another study has performed E. coli RNAtag-seq and Term-seq on the same RNA samples (PMID: 33460557). The authors compare 3' ends identified by Term-seq from this study to other datasets. However, they should also analyze and directly compare their TRS method on the RNAtag-seq in this previous study to the Term-seq 3' end identification in this previous study – as they have done with their own data (Figure 4B).*

Following the reviewer's comment, we conducted comparative analysis for 3' termini determined in the term-seq and RNAtag-seq data of the suggested study (Adams et al., 2021). For both datasets we determined the 3' ends by TRS, as we described in the paper for the comparative analysis of our data. Our analysis has revealed high consistency in the 3' termini determined by TRS using the RNAtag-seq and the term-seq datasets of Adams et al. (2021). 84% of all the 3' termini identified in the term-seq data were identified in the RNAtag-seq data, and ~96% of primary 3' termini identified in the term-seq data were identified in the RNAtag-seq data. These results are mentioned in the main text (page 7 lines 203-207) and described in the Supplemental Information (page 8 lines 193-208).

- *The authors should consider how effective their approach will be for other bacteria. Term-seq has now been carried out for numerous bacteria. This study would be strengthened by analyzing RNAtag-seq data from other bacteria and reporting the efficacy of TRS to find 3' ends. Two studies have recently been submitted to BioRxiv, one for Mycobacterium tuberculosis (<https://doi.org/10.1101/2022.06.01.494293>) and another for Borrelia burgdorferi (<https://doi.org/10.1101/2023.01.04.522626>) which documented a majority of 3' ends mapped internal to ORFs. Will TRS analysis only be successful for a subset of bacterial RNAtag-seq datasets?*

To address the reviewer's comment and assess whether the approach is applicable to other bacteria as well, we conducted two analyses using already published RNAtag-seq data of other bacteria:

- i) For *Listeria monocytogenes*, we had from previously published papers both term-seq data (Dar et al. 2016) and RNAtag-seq data (Avican et al. 2021). We compared 3' termini determined by TRS applied to the RNAtag-seq data and to the term-seq, and showed that 3' termini determined in the two datasets are consistent. The results of the comparative analysis of *Listeria monocytogenes* datasets are described in the revised manuscript (page 7 lines 203-207) and in the Supplemental Information (page 9 lines 212-225, Supplementary Figure 7A-B). We also present the repertoire of 3' termini by their annotations and show their consistency in the two datasets (Supplementary Figure 7C).
- ii) We applied TRS to published RNAtag-seq data of *E. coli* ETEC, *Salmonella enterica* Typhimurium, *Klebsiella pneumoniae*, and *Shigella flexneri* (Avican et al. 2021). We determined the repertoire of 3' termini for each of these organisms, and show that most 3' termini were classified as primary 3' termini. This is described in the revised manuscript (page 7 lines 207-210) and in the Supplemental Information (page 9 lines

225-232, Supplementary Figure 7D). We also present in the revised manuscript a few examples, showing that the positions of 3' termini of orthologous genes are evolutionarily conserved (pages 10-11 lines 307-343 in the manuscript, and Figure 7 of the manuscript).

- *It is not obvious how the last section of the results provides any new insights gained from TRS. Yes, their analysis found several long 3' UTRs from RNAtag-seq data, but these examples (Figure 5) and 3' ends have been reported in other studies, some even with northern analysis to document an sRNA. Rather this section, could speak to novel discoveries from their approach/an example for the application of TRS to find new information from existing datasets.*

We believe that this section is important for multiple reasons, and kept this part in the revised manuscript. First, it demonstrates how combination of multiple layers of information embedded in RNAtag-seq data (read-coverage and 3' termini) can be combined to gain new insights from existing data. Second, we discovered these RNA elements independently and objectively. Having previous experimental evidence in support of some of them, strengthens the validity of the discoveries by our current analysis.

In addition, following the reviewer's comment, we added two new sections to the manuscript, demonstrating the biological knowledge that can be gained by analyzing already available RNAtag-seq data that is stored in public databases. We demonstrate that by applying our algorithm to RNAtag-seq data of enteropathogenic *E. coli* (EPEC) grown under different growth phase/condition, we can identify conditionally-dependent 3' termini that may serve for regulation (pages 9-10 lines 258-305). We exemplify two genes encoding ribosomal proteins for which we identified 3' termini at the 5' UTR or within the coding sequence in growth on LB to stationary phase, but not in growth on DMEM to exponential phase. Our analysis suggests that these genes have shorter RNAs in stationary phase, which we validated experimentally by northern blot analysis (Figure 6C). Our results hint at premature transcription termination or processing in stationary phase of genes encoding ribosomal proteins, reducing their expression level at this growth phase.

In the second section (pages 10-12 lines 307-343), using available RNAtag-seq data, we present the potential of the approach to study regulatory RNA elements from an evolutionary point of view. We analyzed four bacterial species of the Enterobacteriaceae family and show that we can identify regulatory events at the RNA level, which reoccur in these bacteria. Rather than relying solely on sequence conservation, our analysis provides evidence based on experimental data (RNA-seq) that regulatory RNA elements are conserved between bacterial species (Figure 7).

- *The authors should comment on the number of replicates required/ideal for TRS analysis.*

We added our recommendation for the number of replicates (page 19 lines 551-556).

Reviewer #3

This manuscript by Bar et al. presents a bioinformatics method named TRS to identify 3' termini of bacterial transcripts from RNAtag-seq data. The method is based on the hypothesis that real 3' termini of transcripts are significantly enriched in RNAtag-seq data as compared to those artificial termini generated by RNA fragmentation, a step used for transcript sequencing. The methodology is interesting and potentially useful in finding terminus regulation. However, the work as presented is largely a technical trick for terminus analysis. The authors should expand their work to mine existing data in the public database and present termini that are variably used in different conditions. This would make this work suitable for publication in a high profile journal like Nature Communications.

We believe that the approach by itself is a technological advancement, providing a tool for researchers to analyze 3' termini. The approach is not limited to RNAtag-seq data but can be also applied to results generated by other sequencing technologies in which there is enrichment in 3' ends as well (term-seq, for example). We also show in the paper that in case the adapter is ligated at the 5' end, transcription start sites can be identified, and for paired-end sequencing data, both 5' and 3' termini can be identified (page 14 lines 402-410). Thus, highly desired annotations can be derived from available data, making the approach we present very useful.

Following the reviewer's comment, we added two new sections to the manuscript, demonstrating the biological knowledge that can be gained by analyzing already available RNAtag-seq data that is stored in public databases. We also revised the introductory paragraph describing the new insights we demonstrate in the revised manuscript (page 7 lines 212-224). In the first new section we demonstrate that by applying our algorithm to RNAtag-seq data of enteropathogenic *E. coli* (EPEC) grown under different growth phase/condition, we can identify conditionally-dependent 3' termini that may serve for or result from a regulatory event (pages 9-10 lines 258-305). We exemplify two genes encoding ribosomal proteins for which we identified 3' termini at the 5' UTR or within the coding sequence in growth on LB to stationary phase, but not in growth on DMEM to exponential phase. Our analysis suggests that these genes have shorter RNAs in stationary phase, which we validated experimentally by northern blot analysis (Figure 6). Our results hint at premature transcription termination or processing in stationary phase of genes encoding ribosomal proteins, reducing their expression level at this growth phase.

In the second new section (pages 10-12 lines 307-343), using available RNAtag-seq data, we present the potential of the approach to study regulatory RNA elements from an evolutionary point of view. We analyzed four bacterial species of the Enterobacteriaceae family and show that we can identify regulatory events at the RNA level, which reoccur in these bacteria. Rather than relying solely on sequence conservation, our analysis provides evidence based on experimental data (RNA-seq) that regulatory RNA elements are conserved between bacterial species (Figure 7).

We hope that these results, along with the novel 3' UTR-derived small RNAs, discovered by our analysis and presented in the original version of the manuscript (pages 8-9 lines 226-256), make our paper suitable for publication in *Nature Communications*.

In addition, RNA ligation efficiency can have substantial influences on how RNA fragments are captured and made into cDNA. Presumably, some artificial termini generated by RNA

fragmentation may have a much higher ligation efficiency than true termini, leading to false positives. The authors need to address this possibility as well.

Following the reviewer's comment, we conducted a couple of analyses: 1) We aligned the sequences at termini that were determined in the RNAtag-seq data but not in the term-seq data, and analyzed the nucleotide frequencies in flanking positions along the aligned sequences. Indeed, we identified a common sequence motif at these 3' termini, however, it matched the GC-rich sequence of a terminator hairpin structure followed by a poly-uridine tail. This common motif further supports the identified 3' termini as genuine ones. 2) To assess possible ligase preferences, we aligned the sequences at the positions of mapped read starts within CDSs, and looked at the nucleotide enrichment in the flanking regions. There was no nucleotide bias at the ligation point. Interestingly, we identified a slight enrichment of A at the second read position, as well as a slight enrichment of U at the genomic position downstream the mapped read start, possibly associated with some fragmentation bias. Overall, the difference between this motif and the identified terminator motif suggests that the 3' termini identified by TRS are not affected by this slight bias or by any ligase preference. Pages 5-6 lines 148-158, Pages 6-7 lines 178-187 and in the Supplemental Information (Supplementary Fig. 4).

REVIEWERS' COMMENTS

Reviewer #1 (Remarks to the Author):

I am satisfy with the authors' responses.

Reviewer #2 (Remarks to the Author):

The authors have addressed all my concerns and I appreciate their efforts to analyze several additional datasets using “TRS”. Furthermore, the added analyses (conditional termini and regulatory conservation) provide exciting examples for the application of the approach. I recommend the acceptance of the manuscript, which will be a nice contribution to the field.

Reviewer #3 (Remarks to the Author):

The authors have made effort to address my concerns in the revision. I have no more issues.